# Screening and Management of Coronary Artery Disease in Kidney Transplant Candidates

**DOI:** 10.3390/diagnostics13162709

**Published:** 2023-08-20

**Authors:** Giuseppe Vadalà, Chiara Alaimo, Giancarlo Buccheri, Luca Di Fazio, Leandro Di Caccamo, Vincenzo Sucato, Manlio Cipriani, Alfredo Ruggero Galassi

**Affiliations:** 1Division of Cardiology, University Hospital Paolo Giaccone, 90100 Palermo, Italy; odisseos86@gmail.com; 2Department of Health Promotion, Mother and Child Care, Internal Medicine and Medical Specialties (ProMISE), University of Palermo, 90100 Palermo, Italy; chiaraalaimo93@gmail.com (C.A.); giancarlobuccheri92@gmail.com (G.B.); l.difazio93@gmail.com (L.D.F.); leodicaccamo93@gmail.com (L.D.C.); argalassi@gmail.com (A.R.G.); 3Institute of Transplant and Highly Specialized Therapies (ISMETT) of Palermo, 90100 Palermo, Italy; mcipriani@ismett.edu

**Keywords:** kidney transplant, coronary artery disease, end-stage renal disease

## Abstract

Cardiovascular disease (CVD) is a major cause of morbidity and mortality in patients with chronic kidney disease (CKD), especially in end-stage renal disease (ESRD) patients and during the first year after transplantation. For these reasons, and due to the shortage of organs available for transplant, it is of utmost importance to identify patients with a good life expectancy after transplant and minimize the transplant peri-operative risk. Various conditions, such as severe pulmonary diseases, recent myocardial infarction or stroke, and severe aorto-iliac atherosclerosis, need to be ruled out before adding a patient to the transplant waiting list. The effectiveness of systematic coronary artery disease (CAD) treatment before kidney transplant is still debated, and there is no universal screening protocol, not to mention that a nontailored screening could lead to unnecessary invasive procedures and delay or exclude some patients from transplantation. Despite the different clinical guidelines on CAD screening in kidney transplant candidates that exist, up to today, there is no worldwide universal protocol. This review summarizes the key points of cardiovascular risk assessment in renal transplant candidates and faces the role of noninvasive cardiovascular imaging tools and the impact of coronary revascularization versus best medical therapy before kidney transplant on a patient’s cardiovascular outcome.

## 1. Introduction

Cardiovascular disease (CVD) is a major cause of morbidity and mortality in patients with chronic kidney disease (CKD), especially in end-stage renal disease (ESRD) [1]. Furthermore, the risk of cardiovascular adverse events continues to be relevant during the first year after the transplantation, necessitating a regular cardiological follow-up [2,3]. On the other hand, the prevalence of CKD in the general population is increasing [4], especially in some subgroups of patients like diabetics, in which kidney microvascular disease often leads to the need for dialysis or renal transplantation [5]. Furthermore, the prevalence of Type 2 diabetes mellitus (T2DM) has reached pandemic proportions, representing it as a relevant health problem [6]. For these reasons, and due to the shortage of organs available for transplant, it is of utmost importance to identify patients with a good life expectancy after transplant and minimize the transplant peri-operative risk. Severe pulmonary diseases, previous myocardial infarction, or stroke/transient ischemic attack (TIA) within the last 6 months or severe aorto-iliac atherosclerosis are some of the conditions to rule out before listing a patient for kidney transplant [7].

Finally, whether a systematic coronary artery disease (CAD) treatment before kidney transplant may improve short- and long-term cardiovascular outcomes is still under debate, not to mention that a nontailored screening could lead to unnecessary invasive procedures and delay or exclude some patients from transplantation unjustifiably [8].

This review summarizes the key points of cardiovascular risk assessment and CAD management of renal transplant candidates, as suggested by the most representative scientific societies [9,10,11,12,13], facing the role of noninvasive cardiovascular imaging tools and the impact of coronary revascularization versus best medical therapy before kidney transplant on a patient’s cardiovascular outcome.

## 2. CAD Presentation and Its Pathophysiology

ESRD patients have cardiovascular mortality twenty times higher than the general population [14]. At least 35% of these patients complain of effort angina or had a previous myocardial infarction (MI) at the time of first medical contact with the nephrologist. However, stable CAD seems to be more common than acute coronary syndromes (ACS) [15].

Among ACS, non-ST-elevation MI is more frequent than ST-elevation MI [16]. Heart failure (HF), arrhythmias, and sudden cardiac death (SCD) are other possible clinical manifestations [17].

Sometimes these patients may complain of “atypical” symptoms or angina equivalents like dyspnea or fatigue [18]. Besides the classic cardiovascular risk factors (CVRF), like hypertension, hypercholesterolemia, diabetes, tobacco use, and family history, it is supposed that in patients with ESRD additional CKD-related factors might play a role in CAD pathophysiology. Furthermore, whilst the correction of CVRF in the general population results in a well-established cardiovascular event reduction [19], among patients with ESRD, this benefit is so far lower [20]. Nontraditional risk factors can be distinguished into CKD/ESRD-related and hemodialysis-related risk factors. They are summarized in Table 1. 

### 2.1. Classic CV Risk Factors

Traditional CVRF are highly prevalent in CKD patients. They contribute not only to the progression of the atherosclerotic process but also to kidney damage, generating a vicious cycle known as cardiorenal syndrome type 4 [21]. Among all CVRF, hypertension (HTN) and diabetes mellitus (DM) play a pivotal role in CKD development and progression. The relationship between HTN and CKD is complex: HTN is, in fact, both the cause and effect of CKD. Moreover, the prevalence of HTN is roughly 84% among CKD patients, and it becomes more common if diabetic nephropathy coexists [22]. Finally, with the decline of eGFR, HTN incidence, prevalence, and severity significantly increase [23]. 

DM is the most common cause of CKD worldwide. Approximately 20% of all adults with DM have an estimated glomerular filtration rate (eGFR) < 60 mL/min/1.73 m^2^ [24]. DM is now globally the leading cause of ESRD, accounting for roughly 33% of all patients on renal replacement therapy (RRT) [25]. 

Both HTN and DM are responsible for macro and microvascular damage, such as, kidney artery stenosis, nephrosclerosis and glomerular dysfunction, and accelerated coronary and peripheral artery disease. In particular, the term “nephrosclerosis” refers to the process of interstitial inflammatory fibrosis that affects the glomerular capillaries and largely contributes to renal function decline [26,27].

Among traditional CVRF, tobacco use should be mentioned as well. A relationship between tobacco use and its adverse effects on kidney function has been highlighted in several observational studies [28,29,30]. A cross-sectional study by Hallan et al. showed an increased risk of CKD incidence and progression in smokers with a high cumulative lifetime cigarette exposure (RR 1.42 for 25–49 pack years, RR 2.05 for > 50 pack years). However, only a few studies were specifically focused on the CKD population [31]. Among these, a prospective study conducted by Wesson et al. deserves mention. The authors enrolled 216 CKD-stage 2 (eGFR 60–89 mL/min/1.73 m^2^ and spot urine albumin-to-creatinine ratio > 200 mg/g), hypertensive, nondiabetic patients to test if smoking cessation could increase the beneficial effects of ACE inhibition (in particular enalapril) and mitigate kidney damage progression [32]. At 5 years, the smokers had lower eGFR compared with quitters and nonsmokers. Moreover, smoking cessation attenuated eGFR decline compared with smokers (−1.7 ± 1.5 vs. −3.4 ± 1.8 mL/ min/1.73 m^2^/year), while eGFR did not significantly change in nonsmokers (–1.3 ± 1.5 mL/min/1.73 m^2^/year). Different pathogenic mechanisms have been supposed to explain the relationship between cigarette smoking and CKD. Endothelial damage and oxidative stress play a primary role. Furthermore, tobacco use can induce a hyperoxidative stress state in synergy with DM and HTN [33], increasing tissue angiotensin II (AII) level, which exacerbates kidney oxidative stress further [34].

For all these reasons, adequate blood pressure and glycemic control and smoking cessation are strictly recommended, both leading to a reduction in the combined outcome of major adverse cardiac and cerebrovascular events (MACCEs) [35,36].

### 2.2. Anemia

Anemia is a common complication of CKD; it is associated with a reduced quality of life, worse renal survival, and a higher rate of adverse cardiovascular events [37]. Its prevalence and severity are higher as the glomerular filtration declines (ranging between 8.4% at stage I and 53.4% at ESRD). In most cases, CKD-related anemia is due to several complex pathogenic mechanisms, like (1) erythropoietin (EPO) deficiency; (2) iron deficiency due to blood loss or increased hepcidin levels; (3) reduced red cell life span; (4) inflammation and comorbidities; (5) reduced marrow response to EPO; (6) B12 and/or folic acid deficiencies. Among these factors, EPO deficiency is the most important. It is possible to find low EPO levels at the early stages of CKD, but the deficiency becomes clinically relevant when eGFR is <30 mL/min/1.73 m^2^. Chronic low oxygen arterial content due to anemia induces a cardiovascular compensatory response with increased cardiac output and consequent left ventricular hypertrophy (LVH) [38].

### 2.3. Volume Overload, Fluid Retention, and LVH

Sodium loading related to reduced glomerular filtration velocity causes significant fluid retention and consequent volume overload. The main effect of volume overload is the left ventricular mass increase [39].

LVH is present in about one-third of all patients with CKD, and it is an independent predictor of survival in patients with CKD [40]. Furthermore, LVH is one of the most important causes of myocardial ischemia due to the oxygen supply−demand mismatch [41]. LVH in CKD is caused by cardiac preload increase and cardiac afterload boost secondary to a systemic arterial resistance increase. This condition causes cardiac maladaptive changes and cardiomyocyte death, which in turn result in left ventricular concentric remodeling; over time, left ventricle dilatation and systolic dysfunction occur. Moreover, many nonhemodynamic factors also contribute to the development of LVH and cardiomyopathy in CKD patients. Among these, inappropriate activation of the renin–angiotensin system (RAAS), oxidative stress, inflammation, and stimulation of profibrogenic factors (cardiotrophin-1, galectin-3, transforming growth factor-b, fibroblast growth factor-23) deserve mention [42].

### 2.4. Secondary Hyperparathyroidism

CKD is associated with the development of secondary hyperparathyroidism. This condition compromises calcium, phosphate, and vitamin D homeostasis, finally leading to vascular and valvular calcifications. These findings are the result of vascular smooth muscle cell transformation into osteoblast-like cells by the uptake of phosphorus. Phosphate levels and vascular calcification have been associated with increased CAD and cardiovascular events [43].

### 2.5. Endothelial Dysfunction, Oxidative Stress, and Inflammation

CKD causes a chronic pro-inflammatory and pro-oxidative state with immune cell dysregulation, reactive oxygen species (ROS) production, and reduced nitric oxide (NO) availability that, in turn, lead to endothelial dysfunction, vascular remodeling, accelerated atherosclerosis, and myocardial damage [44,45,46].

Patients with CKD may have a dysfunction of different immune cells, which is responsible for negative effects on the kidney and cardiovascular system [47]. For example, it has been shown that monocytes have increased expression of macrophage scavenger receptors (CD36) and consequent enhanced vascular uptake of oxidized, low-density lipoprotein (LDL), contributing to the atherosclerotic process [48]. Moreover, monocyte cells showed an aberrant expression of integrin and toll-like receptors (TLRs) with high expressions of genes encoding pro-inflammatory cytokines (like IL-6, IL-1β and TNFα) [49,50]. Pro-inflammatory cytokines are responsible for dendritic cell activation, which in turn promotes T-cell proliferation. T-CD8+ cells can infiltrate the myocardium and produce potent cytokines, including IL-17, IFN-γ, and TNFα [51]. A variety of immune cells can also induce TGFβ production, which is responsible for fibroblast activation and consequent peri-arteriolar and myocardial fibrosis [52].

The endothelial dysfunction not only facilitates coronary atherosclerotic plaque formation, progression, and destabilization but also involves the coronary microcirculatory system [53]. Different studies have demonstrated the relationship between the degree of glomerular filtration impairment and the reduction of coronary flow reserve (CFR) [54]. High ROS levels are associated with both eGFR decline and CFR impairment. Among ROS, asymmetric dimethylarginine (ADMA) has been shown to compromise coronary endothelial function. In fact, ADMA is an endogenous competitive inhibitor of NO synthase (NOS). In turn, decreased endothelial NO production impairs microcirculation both in the kidneys and heart [55]. Moreover, high ADMA levels have also been associated with higher intima−media thickness and cardiovascular (CV) events in ESRD [56].

### 2.6. Platelet Abnormalities

Acquired platelet abnormalities, such as a reduced serotonin content and an altered thrombin-induced release of adenosine triphosphate (ATP), have been identified in CKD patients. These conditions may promote both a prothrombotic and hemorrhagic state [57].

### 2.7. Uremic Toxins

With the decline of renal function, a large group of uremic solutes, normally cleared by the kidneys, accumulate with a detrimental effect on the cardiovascular system [58]. Over 100 uremic toxins have been identified and classified by The European Uremic Toxin Work Group (EUTox). Among these, an important group is represented by the protein-bound uremic toxins, accounting for approximately 25% of all currently identified uremic toxins, like indoxyl sulfate, hippuric acid, and p-cresyl sulfate. Uremic toxins lead to endothelial dysfunction and ROS production, with negative effects on myocardial and endothelial cells [59,60]. Trimethylamine N-oxide (TMAO), a gut-derived, free water-soluble, low-molecular-weight uremic toxin, deserves mention. TMAO has been recently associated with the atherosclerotic process by NLRP3 inflammasome activation and inflammatory interleukins production, like IL-1β and IL-18. NLRP3 increases endothelial permeability, enhancing macrophage and monocyte infiltration into vascular atherosclerotic lesions [61]. Although hemodialysis prolongs the survival of ESRD patients, it does not completely relieve the uremic state, leaving patients with the so-called “residual syndrome” [62].

### 2.8. Hemodialysis-Related Factors

Although hemodialysis (HD) should improve cardiovascular function by fluid overload correction and uremic toxin removal, cardiovascular mortality in ESRD patients continues to be high [63]. Different HD-related risk factors must be mentioned. Artero-venous (A-V) fistula used for HD creates a reduced vascular resistance state leading to a compensatory hyperactivation of RAAS and the sympathetic system, finally resulting in cardiac output increase. Furthermore, HD itself determines hemodynamic stress; dialysis pulls fluid from the intravascular compartment, which in turn draws fluid from the interstitial. Any mismatch in plasma removal and refill rates can lead to rapid volume contraction and consequent hypotension, which is predictive of increased mortality [64]. Moreover, the interdialytic period is associated with fluid retention and overload increase that adds further stress on the cardiovascular system [65]. Despite the fact that HD removes different toxins that may impair platelet function and hemostasis, current hydrophobic dialyzer membranes promote protein deposition (IgG, C3, fibrinogen, etc.) and consequent abnormal activation of complement, coagulation, and inflammation [66]. Finally, HD contributes to endothelial dysfunction by a NO bioactivity reduction and ROS generation [67].

Despite the lack of clear evidence, a systematic pre-operative clinical evaluation of transplant candidates should identify and correct classical and nontraditional CV risk factors. 

## 3. Cardiac Screening before Kidney Transplantation

This section provides an overview of the available recommendations on CV risk stratification of renal transplant candidates from scientific societies, namely the European Renal Best Practice Transplantation (ERBP), the American Heart Association (AHA), the American College of Cardiology Foundation (ACC), and the European Society of Cardiology (ESC) (Table 2). 

Different approaches to prerenal transplant screening of cardiovascular disease have been proposed to reduce the number of unfavorable cardiovascular events occurring at the time of and after kidney transplant. All these protocols are based on the patient’s cardiovascular risk stratification. Cardiac screening has the task of excluding from the waiting list those candidates with high cardiovascular perioperative risk and those with well-known CAD and long-term prognosis concerns [68].

**Table 2 diagnostics-13-02709-t002:** Cardiovascular screening before renal transplant of high cardiovascular risk and asymptomatic patients according to ERBP, the European Renal Best Practice Transplantation; AHA/ACC, American Heart Association/American College of Cardiology; and ESC, European Society of Cardiology.

Society	High-Risk Criteria	Management
AHA/ACC (2012) [12]	Diabetes mellitus Prior CVD More than one year on dialysis LVH Age > 60 years Smoking Hypertension Dyslipidemia	All patient: ECG + echocardiogram High-risk patients: noninvasive stress testing Coronary angiography if positive noninvasive stress test or symptom of CAD.
ERBP (2013) [13]	Older age Diabetes mellitus Prior CVD	All patients: ECG + chest x-ray High-risk patients: exercise tolerance test and echocardiogram Noninvasive stress imaging if positive or inconclusive exercise tolerance test Coronary angiography if positive test for cardiac ischemia or symptom of CAD
AHA (2022) [69]	Age > 60 years Diabetes mellitus Smoking Prior cerebrovascular disease PAD Duration dialysis + prior kidney transplantation > 5 years	No prior CAD: All patients: ECG + echocardiogram High-risk patients: functional or anatomic assessment Coronary angiography if ECG, echocardiogram, functional of anatomic assessment positive or symptom of CAD Prior CAD: Coronary angiography >2 years: functional assessment Coronary angiography <2 years: no revascularized CAD? New coronary angiography if functional assessment is positive or previous CAD not revascularized
ESC (2022) [70]	Age > 65 years Smoking Hypertension Diabetes Dyslipidemia Family history of CVD	All patients: history + physical examination + standard laboratory test High-risk patients/Prior CAD: ECG + biomarkers + functional capacity If ECG or symptoms suggestive of CAD → echocardiogram → stress imaging → coronary angiography

ERPB = European Renal Best Practice Transplantation; AHA/ACC = American Heart Association/American College of Cardiology; ESC = European Society of Cardiology; ECG = electrocardiogram; LVH = left ventricular hypertrophy; CAD = coronary artery disease; CVD = cardiovascular disease; PAD = Peripheral artery disease.

While symptomatic patients for angina or angina equivalents should be referred to a cardiologist to evaluate the need for myocardial revascularization following reference guidelines on chronic coronary syndromes in ESRD, the role of screening in asymptomatic/low-risk CAD patients is less obvious [9,69]. Given that transplant candidates often had significant CAD without symptoms, the enhancement of pre-transplant screening should, in theory, improve outcomes, but this is not always the case.

Indeed, in a national observational propensity cohort study, among 2572 kidney transplant candidates asymptomatic for CAD, 50% underwent CAD screening by stress test, coronary CT, and/or coronary angiogram, while the remaining 50% did not. The study showed that the risk of MACE at 90 days (HR 0.80, 95% CI 0.31–2.05; *p* = 0.64), at one year (HR 1.12, 95% CI 0.51–2.47; *p* = 0.77), at five years (HR 1.31, 95% CI 0.86–1.99; *p* = 0.20) and after transplantation, was similar in both groups [71].

### 3.1. Asymptomatic Patient—Low Cardiovascular Risk

The 2013 European Renal Best Practice Transplantation (ERBP) Guidelines suggested that, besides clinical data collection, the physical examination, resting electrocardiogram (ECG), and chest X-ray are sufficient to rule out CAD in asymptomatic low cardiovascular risk kidney transplant candidates (younger age, no diabetes, no history of cardiovascular disease) [13].

Similarly, the American Heart Association (AHA) and the American College of Cardiology Foundation (ACC) recommend an approach comprehensive of the patient’s clinical history, a thorough physical examination, a resting 12-lead ECG, and left ventricular ejection fraction (LVEF) assessment through echocardiography (Figure 1) [12,69].

According to the recent European Society of Cardiology (ESC) Guidelines on “Cardiovascular Assessment and Management of Patients undergoing Non-Cardiac Surgery (NCS)” (Figure 1), the renal transplant is listed among surgery at intermediate risk of 30-day MACE (1–5%). As such, all patients scheduled for kidney transplantation should undergo an accurate medical history and physical examination. Patients <65 years, who have no signs, symptoms, or a history of CVD or CV risk factors, are deemed low risk and can undergo moderate-risk surgery without an extra pre-operative risk evaluation. However, a 12-lead ECG and transthoracic echocardiography (TTE) before NCS in individuals with a family history of hereditary cardiomyopathy, regardless of age or symptoms, is recommended [70].

In conclusion, American and European scientific societies agree on a conservative screening approach for asymptomatic kidney transplant candidates with low cardiovascular risk. Stress tests and/or coronary angiography are not necessary if initial screening tests yield negative results [12,13,69,70].

### 3.2. Asymptomatic Patient—High Cardiovascular Risk

Table 2 shows the high cardiovascular risk features proposed by American and European scientific societies.

The ERBP Guideline recommends an echocardiography and an exercise tolerance test for all patients, while a stress imaging test (stress echocardiography (SE) or myocardial perfusion studies (MPS) is suggested only in cases of positive or inconclusive preliminary exams. Finally, coronary angiography is indicated if noninvasive stress imaging is positive [13]. SE sensitivity and specificity in CKD patients are 0.73 and 0.88, respectively, whereas MPS sensitivity and specificity are 0.66 and 0.75, respectively [69]. Because of increased arterial calcification in CKD, the efficacy of coronary computed tomography angiography (CCTA) to define the degree of obstructive lesions is limited (sensitivity 0.96 and specificity 0.66). A normal CCTA may offer reassurance, but an abnormal result could be of nonunivocal interpretation because of a 29% false-positive [12].

According to the 2022 Scientific Statement from the AHA (Figure 2), when a patient with no history of CAD has a high cardiovascular risk, functional testing (SE or MPS) or anatomic testing (CCTA or invasive coronary angiography [CA]) is recommended. Differently, if a patient has a known CAD, it is advisable to evaluate their most recent coronary angiography. At the same time, if the previous CA was conducted more than 2 years before, a stress test is recommended before further investigations [69].

The ESC Guideline (Figure 2) considers people < 65 years, as well as those who have at least one risk factor for CVD, to be at a greater risk of having undetected CVD. These individuals require additional evaluation prior to intermediate-risk surgery, such as kidney transplant, as well as appropriate management of risk factors. ECG, functional capacity assessment (self-reported ability to climb two flights of stairs), and/or biomarker measurement (high-sensitivity cardiac troponin T or I [hs-cTn T or hs-cTn I] and/or N-terminal pro-B-type natriuretic peptide [NT-proBNP]/B-type natriuretic peptide [BNP]) are all indicated. It is recommended to measure hs-cTn T or hs-cTn I before surgery, as well as 24 and 48 h thereafter. In patients with known CAD, the same process should be followed [70]. Trans thoracic echocardiography (TTE) may be considered before kidney transplant in patients with low functional capacity, abnormal ECG, high NT-proBNP/BNP, or ≥1 clinical risk factor. When ischemia is a concern in individuals with clinical risk factors and low functional capacity, stress imaging may be considered before intermediate-risk NCS. CCTA should be considered to rule out CAD in patients with suspected chronic coronary syndromes (CCS) or biomarker-negative non-ST elevation acute coronary syndromes (NSTE-ACS) in case of low-to-intermediate clinical likelihood of CAD, or in patients unsuitable for noninvasive functional testing undergoing nonurgent, intermediate-risk NCS [70].

## 4. Revascularization vs. Best Medical Therapy (BMT) before Transplantation: A Question Still Unanswered

CAD management in kidney transplant candidates is cumbersome. The main indication for revascularization is driven by the persistence of symptoms despite optimized medical therapy (OMT) and/or to improve prognosis.

### 4.1. Optimized Medical Therapy

The management of CAD in CKD patients should be the same as in the general population [42,72]. Nonpharmacological treatment, such as lifestyle optimization, smoking cessation, and weight loss should always be recommended, according to current guidelines for CV prevention [19].

The use of antiplatelets, statins, angiotensin-converting enzyme inhibitors (ACE-I), angiotensin-receptor blockers (ARB), and beta-blockers are currently recommended. The use of these drugs in waiting list patients, during the peri-operative period and late after kidney transplant guidelines is also recommended [73]. However, underutilization of these drugs in CKD patients is caused by some concerns about the bleeding risk or the worsening of renal function. About this topic, Qiao et al. highlighted that interruption of ACE-I or ARB therapy in patients with decreasing renal function is associated with a higher risk of mortality and MACE [74]. While β-blocker treatment reduced the risk of all-cause (RR: 0.72, 95% CI: 0.64–0.80) and cardiovascular mortality (RR: 0.66, 95% CI: 0.49–0.89) in all CKD stages [75], ACE-I and ARB have been shown to reduce morbidity and mortality in several large randomized trials [76], but few data are available on patients with advanced CKD [77].

#### 4.1.1. Lipid-Lowering Drugs

The Study of Heart and Renal Protection (SHARP) examined the effect of simvastatin versus simvastatin plus ezetimibe in patients with advanced chronic kidney disease without a previous coronary artery disease. The study demonstrated that lowering LDL cholesterol with simvastatin plus ezetimibe produces a 17% reduction in the primary endpoint of cardiovascular death, nonfatal myocardial infarction, nonfatal stroke, or coronary revascularization [78].

Similarly, Sooho et al., in late-stage chronic kidney disease in a postend-stage renal disease transition population, showed that statin therapy reduces the all-cause of death and cardiovascular mortality at twelve months, as well as the hospitalization rate after initiating dialysis compared to the group of patients not treated by statins. Over 12 months of follow-up, patients who received statin therapy in the year before the transition to end-stage renal disease had a lower all-cause mortality rate (33.1 [95% CI, 32.3–33.8] vs. 37.9 [95% CI, 37.0–38.8] per 100 person-years, respectively), a 12% lower all-cause mortality risk in the unadjusted model (HR, 0.88 [95% CI, 0.85–0.91]), and (in adjusted models) a 17% lower risk of cardiovascular death (HR, 0.83 [95% CI, 0.78–0.88]) if compared with those who did not receive statin therapy [79].

#### 4.1.2. Upcoming Role of Gliflozines

In recent years, sodium−glucose cotransporter 2 inhibitors (SGLT2i) have been shown to decrease both kidney and cardiovascular adverse outcomes in patients with kidney disease and heart failure [80,81,82,83]. Furthermore, the CREDENCE trial and the DAPA-CKD trial evaluated the efficacy of SGLT2 inhibitor therapy in patients with CKD, with and without Type 2 diabetes [84,85].

In the CREDENCE trial, the event rate of the composite of end-stage kidney disease, doubling of the serum creatinine level, or renal or cardiovascular death was significantly lower in the canagliflozin group than in the placebo group (43.2 and 61.2 events per 1000 patient-years, respectively), which resulted in a 30% relative risk reduction [HR = 0.70; 95%CI = 0.59–0.82; *p* = 0.00001] [84]. Similarly, the DAPA-CKD showed a significant risk reduction of the composite of sustained decline in the estimated GFR of at least 50%, end-stage kidney disease, or death from renal causes [OR = 0.56; 95%CI = 0.45–0.68; *p* < 0.001)], and of the composite of death from cardiovascular causes or hospitalization for heart failure [HR = 0.71; 95%CI = 0.55–0.92; *p* = 0.009)]. Of note, the DAPA-CKD trial showed an all-cause mortality risk reduction [HR = 0.69; 95% CI = 0.53–0.88; *p* = 0.004] [85,86].

However, data about the safety and efficacy data on SGLT2i therapy in kidney transplant recipients are very limited because these studies focused predominantly on short-term outcomes, like glycemic control, body weight reduction, eGFR, and blood pressure (BP) changes. Long-term outcomes on chronic allograft function, cardiovascular morbidity and mortality, as well as graft and patient survival, remain to be explored in the future [87].

### 4.2. Revascularization

Although a large number of previous randomized trials and meta-analyses, showed that myocardial revascularization of symptomatic patients with high-risk CAD had a better outcome compared with those treated by medical therapy only, these studies were not specifically designed for patients with end-stage renal failure on dialysis and/or waiting for renal transplantation [88,89,90,91,92].

On the other hand, the KDIGO Clinical Practice Guideline on the Evaluation and Management of Candidates for Kidney Transplantation recommends excluding from the kidney transplant waiting list those asymptomatic patients with severe CAD unless their estimated survival meets the acceptable criteria outlined by national standards and in selected cases after successful revascularization (Figure 3) [9]. Beyond this background, the role of myocardial revascularization before kidney transplant is still a question without a definitive answer, and the available evidence leads to conflicting results.

Kumar et al. evaluated the role of coronary angiography in all potential transplant candidates over 50 years, and/or with diabetes, cardiovascular disease, symptoms or ECG signs of ischemia or previous myocardial infarction. Myocardial revascularization was performed in 28% of patients. At one and three-year follow-ups, the cardiac event-free survival was respectively 98.0% and 88.4% in those patients who underwent transplant after revascularization, and 94.0% and 90.0% in those on dialysis treatment and still on the waiting list [93].

Bangalore et al. compared a revascularization strategy by CABG or PCI in a cohort of 5920 patients with chronic renal failure. Compared with the CABG group, the PCI group had a lower short-term risk of death (1.0% vs. 1.7%; *p* = 0.01), stroke (0.4% vs. 1.7%; *p* < 0.0001), and lower risk of repeated revascularization (0.4% vs. 0.8%; *p* = 0.04). Conversely, at four years follow-up, in the subgroup of patients on dialysis therapy, PCI had higher death (54.3% vs. 39.1%; *p* = 0.0002), MI (31.9% vs. 16.7%; *p* = 0.05), and repeated revascularization (48.3% vs. 25.0%; *p* = 0.0003) rates than CABG [94].

Finally, the recent ISCHEMIA-CKD trial [8] compared the effects of a conservative versus interventional approach in patients with chronic coronary syndrome and advanced kidney disease (defined by eGFR < 30 mL/min/1.73 m^2^ or treated with dialysis therapy). Compared to optimized medical therapy, myocardial revascularization did not show any benefits. 

However, the ISCHEMIA-CKD study was not specifically designed for kidney transplant candidates.

In a posthoc analysis of the ISCHEMIA-CKD study, Herzog et al. focused on the subgroup of patients included in the study who underwent cardiac stress testing prior to randomization as part of their assessment for kidney transplant listing. Among patients assigned to an invasive strategy versus conservative strategy, the adjusted hazard ratios for the primary outcome were 0.91 (95% confidence interval [CI]: 0.54–1.54) and 1.03 (95% CI: 0.78–1.37) for those listed and not listed, respectively (*p* of interaction = 0.68). The authors concluded that an invasive strategy in kidney transplant candidates did not improve outcomes compared with conservative management [95]. 

In a recent meta-analysis, Siddiqui et al. evaluated the potential benefit of coronary revascularization over optimal medical therapy in renal transplant candidates with asymptomatic coronary artery disease and heterogeneity of characteristics and risk factors. The primary study endpoints were all-cause mortality, cardiovascular mortality, and MACE. Eight studies comprising 945 patients were included. The meta-analysis showed no difference in all-cause mortality, cardiovascular mortality, or MACE rate between candidates who underwent revascularization and those managed by OMT [96].

In conclusion, the available studies did not support routine coronary angiography or revascularization in all patients with advanced CKD and chronic coronary syndromes listed for transplant. 

#### Dual Antiplatelet Therapy (DAPT) Duration after PCI

The need for DAPT after percutaneous revascularization is one of the main causes of kidney transplant delay. Furthermore, ESRD patients have a high hemorrhagic risk and a concomitant prothrombotic state, making their management very complex [92]. The guidelines on percutaneous coronary revascularization do not provide specific indications on antiplatelet therapy duration in kidney transplant candidates, and very few data on this subset of patients are available [73,92].

Dogan et al. investigated the safety of an early DAPT discontinuation (Aspirin plus Clopidogrel) in renal transplant waiting list patients who underwent a second-generation drug-eluting stent (DES) implant or a bare-metal stent (BMS). According to DAPT duration, the DES group was divided into DES early interruption (three months after DES implantation) and DES late interruption (3–12 months after DES implantation); in the bare-metal stent group, DAPT duration was at least one month. No difference in myocardial infarction, death, and MACEs was found between DES-early and DES-late groups [97]. For this reason, a three-month DAPT therapy after a second-generation DES implant, is safe and does not increase myocardial infarction and MACE rates in kidney transplant candidates. Similarly, the latest generation DES showed a good safety profile with a one-month duration of DAPT [98,99,100]. In conclusion, early DAPT interruption after new generation DES seems to be safe even in transplant candidates. 

## 5. Future Perspectives

One of the main questions still unanswered is if a systematic CAD screening among asymptomatic kidney wait-list patients can definitively improve patients’ outcomes. An ongoing randomized controlled trial named the CARSK study (Canadian Australasian Randomized Trial of Screening Kidney Transplant Candidates for CAD) will test the hypothesis that eliminating screening tests for unknown CAD is not inferior to regular screening for the prevention of MACEs of waiting list patients. By validating or refuting the use of screening tests during wait listing, CARSK will ensure judicious use of health resources and optimal patient outcomes [101].

There is a growing use of noninvasive imaging assessment of the coronary anatomy, such as cardiac magnetic resonance (CMR) and positron emission tomography (PET), but more research is needed to precisely evaluate the sensitivity and specificity of these tests in patients in kidney transplant wait lists [102].

## 6. Conclusions

Cardiovascular disease is the leading cause of morbidity and death in CKD patients. An accurate cardiovascular risk assessment of transplant candidates is mandatory to properly allocate the few available organs to those patients who have a low peri-operative transplant risk and a reasonable long-term life expectancy. However, in those more controversial cases, a multidisciplinary approach might be necessary to decide if a candidate must be maintained or excluded from the waiting list.

## Figures and Tables

**Figure 1 diagnostics-13-02709-f001:**
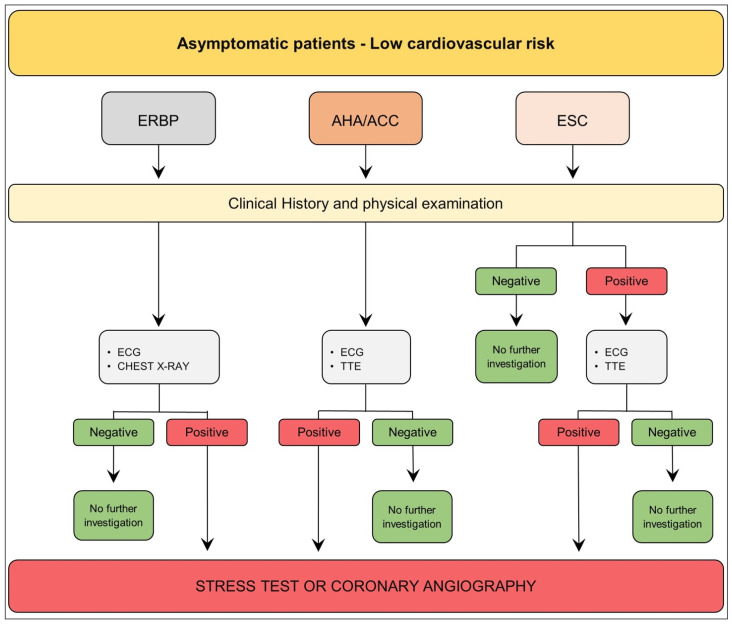
Pre-operative assessment before kidney transplantation of low cardiovascular risk and asymptomatic patient according to ERBP, the European Renal Best Practice Transplantation; AHA/ACC, American Heart Association/American College of Cardiology; ESC, European Society of Cardiology. ERPB = European Renal Best Practice Transplantation; AHA/ACC = American Heart Association/American College of Cardiology; ESC = European Society of Cardiology; ECG = electrocardiogram; TTE = transthoracic echocardiography. Cardiovascular risk factors: hypertension, smoking, dyslipidemia, diabetes, family history of cardiovascular disease.

**Figure 2 diagnostics-13-02709-f002:**
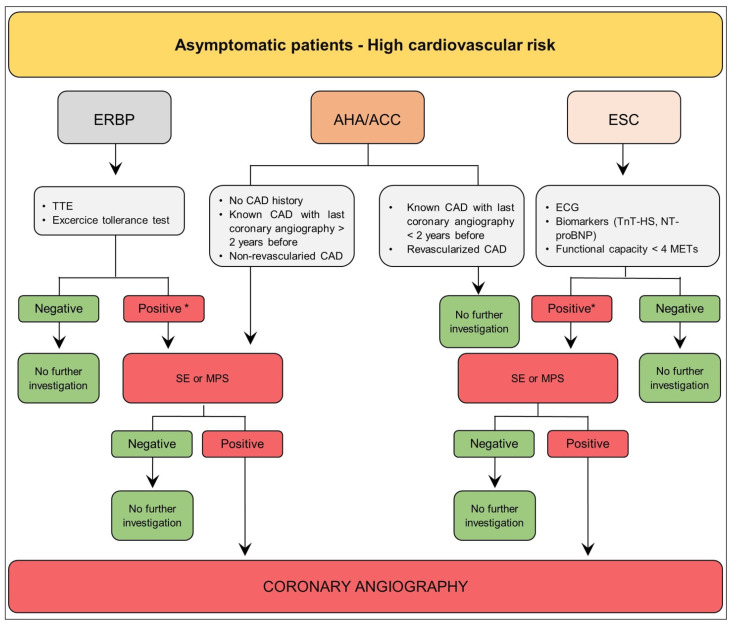
Algorithm for coronary artery disease screening in kidney transplant candidates at high cardiovascular risk and asymptomatic according to the ERBP, the European Renal Best Practice Transplantation; AHA/ACC, American Heart Association/American College of Cardiology; ESC, European Society of Cardiology. ERPB = European Renal Best Practice Transplantation; AHA/ACC = American Heart Association/American College of Cardiology; ESC = European Society of Cardiology; TTE = transthoracic echocardiography; CAD = coronary artery disease; ECG = electrocardiogram; TnT-HS = High-Sensitivity Cardiac Troponin T/I; NT-proBNP = N-terminal prohormone of brain natriuretic peptide; METs = metabolic equivalents; SE = stress echocardiography; MPS = myocardial perfusion studies. Cardiovascular risk factors: hypertension, smoking, dyslipidemia, diabetes, family history of cardiovascular disease. Biomarkers: High-Sensitivity Cardiac Troponin T/I (TnT-HS T/I) and/or Brain Natriuretic Peptide/ N-terminal prohormone of brain natriuretic peptide (BNP/NT-proBNP). (*) at least one.

**Figure 3 diagnostics-13-02709-f003:**
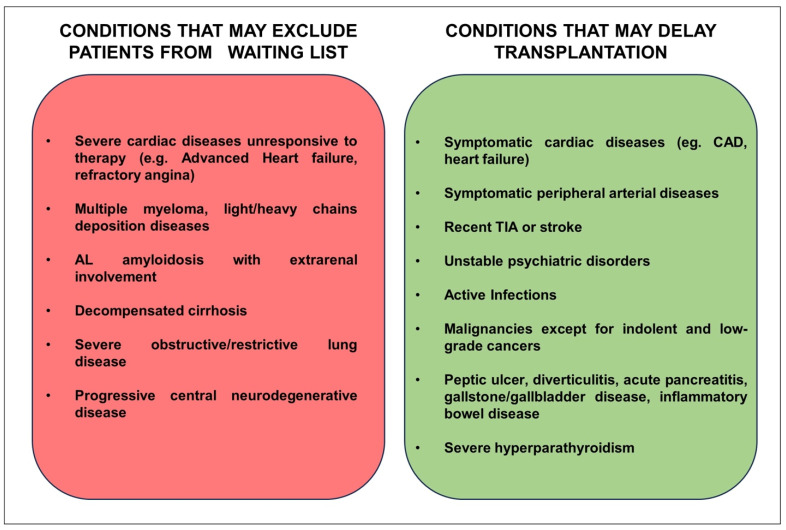
Conditions that can exclude or postpone a candidate from the waiting list.

**Table 1 diagnostics-13-02709-t001:** Traditional and nontraditional risk factors involved in CV risk of ESRD patients. ROS = reactive oxygen species. A-V = artero-venous.

Traditional Risk Factors	CKD-Related Risk Factors	Haemodialysis-Related Risk Factors
Age	Inflammation and oxidative stress (ROS)	A-V fistula
Sex	Anemia	Biomaterials contact
Family history	Uremic toxins	Hemodynamic stress
Hypertension	Hypervolemia	Hemoglobin decompartmentalization
Diabetes mellitus	Secondary hyperparathyroidism	
Dyslipidemia	Prothrombotic state and platelet dysfunction	
Tobacco use		
Physical inactivity		

## Data Availability

No new data were created.

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
