# Peer review of "Screening and Management of Coronary Artery Disease in Kidney Transplant Candidates"

_diagnostics, 2023, doi:10.3390/diagnostics13162709_

Round 1

Reviewer 1 Report

The manuscript entitled, “Screening and management of coronary artery disease in kidney transplant candidates" by Vadalà et al., summarized the key points of cardiovascular risk assessment in renal transplant candidates. This review discussed on various aspect of screening and management of coronary artery disease in kidney transplant candidates. Transplant organs must be distributed fairly among patients with low perioperative transplant risk and reasonable long-term life expectancy; hence a reliable cardiovascular risk assessment of transplant candidates is essential. Overall, this is a well written, significant and well-timed article, this reviewer has certain recommendations that would assist to produce a more comprehensive overview of the topic: 

Comments:

1, The English of manuscript can be polished (minor).

2, The authors should cross-check all abbreviations in the manuscript. Initially, define in full name followed by abbreviation.

3, It will be interesting to add immune cells prospective to this study as immune cells plays vital role in cardiovascular diseases (PMID: 16751419, PMID: 36093172, PMID: 36465455, PMID: 35730443, PMID: 36337927; PMID: 34630414; PMID: 34043424; 34119620 etc).  

5, Authors can include the future directions to their study. 

6, At least one illustrative figure may be provided as to highlight the summary of this study.

Minor editing of English language required

Reviewer 2 Report

Dear authors, 

Thank you for drafting a review on the topic of screening and management of CAD in CKD patients prior to kidney transplantation. 

Overall, the text is easy to read and picks up on the main professional societies. However, I lack the methodological indication of how the literature was selected and why these particular professional societies are mentioned. In addition, it remains unclear to me, for example, why smoking is not really mentioned in point 2 in line 55 and why no further study data are mentioned. All in all, further statistical data in the first part would also be a clear benefit for the publication. Likewise, the question would be whether the aspect of endothelial dysfunction as well as the determination of arterial stiffness as a non-invasive measurement is not really mentioned. In management, the topic of smoking cessation is also not mentioned, this should also be supplemented.

Dear authors,

thank you very much for your review. With regard to the English language, I would like to note that many spaces used twice in a row appear in the reading flow and the publication should be looked at again for these duplications.

Reviewer 3 Report

Vadalà et al. reviewed screening and management of coronary artery disease in kidney transplant candidates. This is an important topic given the limited number of organs available for transplantation and several international recommendations exist. They are summarized in a table in the current review. Risk factors are also summarized and several strategies for the management of coronary artery disease described and discussed including the underlying available studies. The manuscript is overall informative. The introduction part should be re-written as it resembles the abstract.

Quality of English is good, some errors in grammar should be corrected.

Round 2

Reviewer 2 Report

Dear Ladies and Gentlemen,

Thank you very much for revising the publication. From my point of view, the work has clearly won and there is nothing to be said against the publication.